# Magnetic-Free Isolators Based on Time-Varying Transmission Lines

**Fengchuan Wu [1], Yuejun Zheng [2], Fang Yuan [2] and Yunqi Fu [1,\*]**

[1]  College of Electronic Science and Technology, National University of Defense Technology, Changsha 410073, China; wufengchuan17@nudt.edu.cn

[2]  Information and Navigation College, Air Force Engineering University, Xi'an 710043, China; erikzhengyang@126.com (Y.Z.); 13379260913@163.com (F.Y.)

[\*]  Correspondence: yunqifu@nudt.edu.cn

**Abstract:** Two magnetic-free reconfigurable isolators based on a doubly balanced gyrator (DBG) are designed in this paper. One of the isolators is a single-ended transmission isolator (STI), which uses two matching resistors to absorb the signal transmitted in the reverse direction. In theory, it has infinite isolation bandwidth, which is verified by simulation and an experiment. The other isolator is a differential transmission isolator (DTI) to improve the anti-interference performance, which consists of four Wilkinson power splitters (combiners) and two reciprocal transmission line segments. The DTI uses two pairs of differential signals to prevent the reverse signal. Compared to the STI, the DTI has higher power capacity. Furthermore, when the phases of the control signals acting on the switches are changed, the isolation directions of the two isolators will be changed, to obtain the reconfigurable property.

**Keywords:** isolators; magnetic-free; non-reciprocal; reconfigurable; time-varying transmission line

## 1. Introduction

Non-reciprocal components are essential to many modern communication and sensing systems in realizing many important functions that cannot be replaced by reciprocal devices. Traditional non-reciprocal devices such as circulators and isolators use ferrite materials with external magnetic field bias to realize non-reciprocal properties [1,2]. However, the preparation technology of ferrite material is not compatible with the traditional integrated technology. The integration of non-reciprocal devices has been a challenge for decades. Therefore, research for magnetic-free properties has become a new way to solve the problem. The ability to integrate magnetic-free non-reciprocal components into modern semiconductor processes would enable exciting frontiers for constructing communication and sensing systems.

Magnetic-free circulators based on spatiotemporally modulated rings are proposed by Andrea Alù et al. [3–5]. The use of localized resonances reduces the size, but there is a limitation in the bandwidth of operation. Furthermore, it is hard to extend this technique to millimeter waves.

In recent years, magnetic-free properties based on time-varying transmission lines (TVTLs) have provided some new methods for the integration of non-reciprocal devices. Shihan Qin et al. proposed one kind of TVTL to realize magnetic-free non-reciprocal devices [6,7], where the distributed capacitive mixers are used in the TVTLs to construct distributed parametric effect. When the direction of the transmission signal is the same as the carrier's, the transmission signal will mix with the carrier. However, if the transmission signal and the carrier are in opposite directions, the two signals will not mix up. Therefore, the TVTL can be used as a frequency conversion isolator or a circulator, or even a traveling wave amplifier [8]. However, a lengthy circuit is needed to realize spectrum shift. A non-reciprocal

phase shifter based on staggered commutation is proposed by Negar Reiskarimian et al., which consists of an N-path filter and two CMOS time-varying switch sets with asymmetrical control signals [9,10]. The non-reciprocal phase shifter was used to design a magnetic-free circulator, which was integrated on a chip for the first time. However, this approach suffers from low bandwidth, because the capacitors in the N-path filters do not provide a true time delay.

Dinc et al. proposed two kinds of phase shifters in 2017 [11,12], which used switch-controlled TVTL to realize a balanced gyrator (BG) and a doubly balanced gyrator (DBG). In 2018, they further proposed two kinds of isolators based on the BG and the DBG [13]. However, the isolator based on the BG, called the "ultra-broadband dissipative isolator (UDI)", cannot effectively suppress the voltage swing caused by the switch, and consequentially has limited power capacity and isolation. The isolator based on the DBG could shift the spectrum of the reverse signal to prevent reception of the frequency from the transmitter, which is called the "frequency conversion isolator (FCI)". The FCI has a reconfigurable property, but a limited isolation bandwidth.

In this paper, two kinds of magnetic-free isolators based on the DBG are proposed. The single-ended transmission isolator (STI) has ultra-broadband and reconfigurable properties, which have not been realized at the same time by the FCI and the UDI. However, it has limited anti-interference and power capacity ability. So, the differential transmission isolator (DTI) is proposed to solve these problems, which uses differential transmission structure to improve the anti-interference performance. It uses two pairs of differential signals to realize the isolator, which is different from the FCI and the UDI in principle. Moreover, it has larger power capacity ability by using Wilkinson power splitters (combiners) to protect the circuit better. The broadband property of the DTI is verified by using the simplest structure of Wilkinson power splitters (combiners). The bandwidth of the DTI can be expanded by using broadband power splitters (combiners). Besides, it also has a reconfigurable property.

This paper is organized as follows: Section 2 describes the STI design and the simulation work done to verify the broadband isolation property. Section 3 describes the DTI, which can improve the anti-interference performance and power capacity, and the simulation that was done to verify the broadband property and anti-interference. Section 4 presents experiments designed to verify the ultra-broadband and isolation properties of the STI, and discusses the experimental error. Finally, a conclusion is made in Section 5.

## 2. The Single-Ended Transmission Isolator

The schematic of the DBG and the switch control signals are shown in Figure 1a, which consists of two Gilbert-quad switch sets (GQSS) with two transmission line segments sandwiched between them. This has been introduced in a previous work [11]. Here, the Ts represents the periods of the clock signals. The clock signals of the right switches are delayed with respect to the left switches by a value which equals to the propagation delay of the transmission line Ts/4. Assuming that the DBG is a well-matched four-port network, the scattering parameter matrix can be written as:

$$S = \begin{pmatrix} 0 & 0 & 0 & e^{-j\omega\frac{T_s}{4}} \\ e^{-j\omega\frac{T_s}{4}} & 0 & 0 & 0 \\ 0 & e^{-j\omega\frac{T_s}{4}} & 0 & 0 \\ 0 & 0 & e^{-j\omega\frac{T_s}{4}} & 0 \end{pmatrix},\tag{1}$$

Here, $\omega$ represents the transmission frequency. The transmission direction is 1-2-3-4-1, according to Equation (1). If port 3 and port 4 are connected with matching loads, as shown in Figure 1b, this structure will degenerate to a two-port network. The scattering parameter matrix can be written as:

$$S = \begin{pmatrix} 0 & 0 \\ e^{-j\omega\frac{T_s}{4}} & 0 \end{pmatrix},\tag{2}$$

So, a frequency-independent isolator can be constructed, which is called the STI. Assuming that the loads at port 3 or port 4 cannot totally absorb the reverse signal, the signal that has not been absorbed will be transmitted to the next port connected with the load, where it will be further absorbed. As a result, the isolator has enough isolation to isolate the reverse signal.

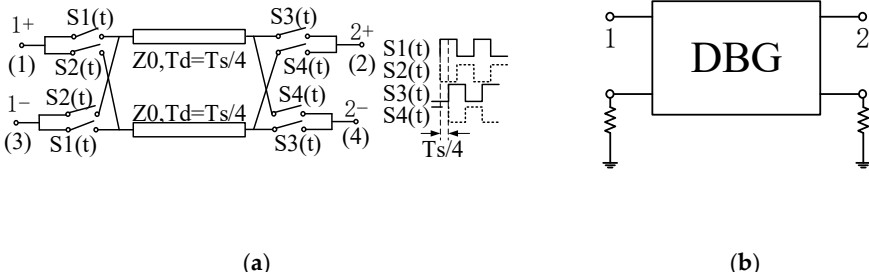

(**a**)                                                                    (**b**)

**Figure 1.** (**a**) Schematic of the doubly balanced gyrator (DBG); and (**b**) schematic of the single-ended transmission isolator (STI).

The Harmonic Balance module in the Keysight Advanced Design System (ADS) simulation software was used to get the spectrum to verify the theory. Assuming that all the transmission line segments with no dispersion have a delay time of Ts/4 = 0.5 ns, the phase relationships between the clock signals are the same as shown in Figure 1a. We defined the frequency range of the input signal to be from 0.1 to 1 GHz, with a step size of 0.1 GHz. The input power was set as 10 dBm. Figure 2a shows the simulation results when the signal inputs from port 1. The results indicate that the signal can pass the isolator almost without loss. However, there was a large loss when the signal was input from port 2, as shown in Figure 2b. The isolation was about 60 dB throughout the simulation band.

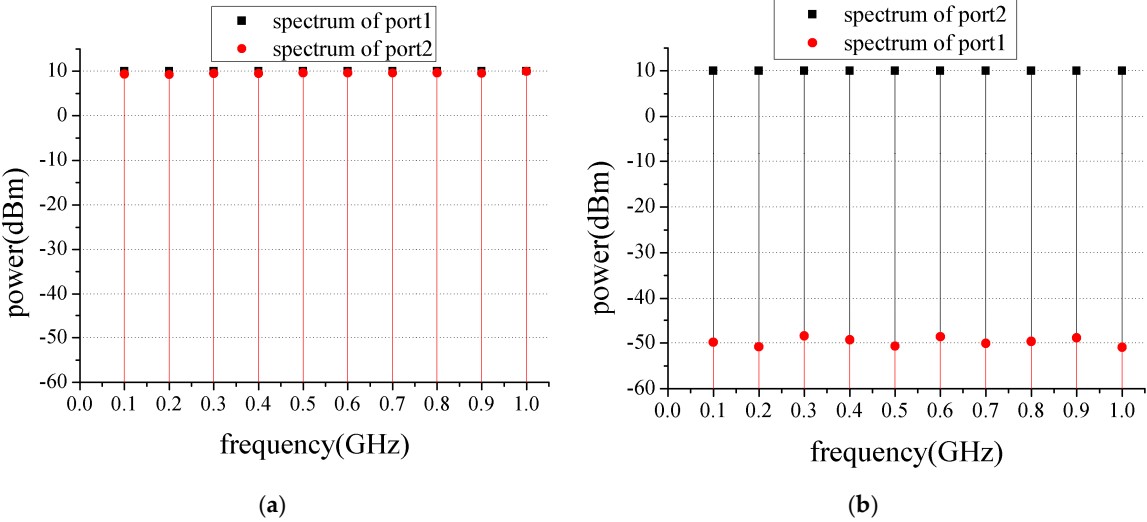

(**a**)                                                                    (**b**)

**Figure 2.** Spectrum analyzing results of the STI. (**a**) Input from port 1; and (**b**) input from port 2.

## 3. The Differential Transmission Isolator

Since the STI has limited power capacity and is also easily affected by the interference, the DTI was designed, which uses two pairs of differential signals to prevent reverse signals. It has better power capacity ability and anti-interference performance compared to the STI. When the DBG is used as the differential transmission structure, the scattering matrix of the DBG can be written as:

$$S = \begin{pmatrix} 0 & -e^{-j\omega \frac{T_s}{4}} \\ e^{-j\omega \frac{T_s}{4}} & 0 \end{pmatrix},$$

(3)

The S parameter matrix of the reciprocal transmission line segment, with a delay time of Ts/4, can be written as:

$$S = \begin{pmatrix} 0 & e^{-j\omega\frac{T_s}{4}} \\ e^{-j\omega\frac{T_s}{4}} & 0 \end{pmatrix}, \tag{4}$$

Adding Equation (3) and Equation (4), the S parameter matrix can be written as:

$$S = \begin{pmatrix} 0 & -e^{-j\omega\frac{T_s}{4}} \\ e^{-j\omega\frac{T_s}{4}} & 0 \end{pmatrix} + \begin{pmatrix} 0 & e^{-j\omega\frac{T_s}{4}} \\ e^{-j\omega\frac{T_s}{4}} & 0 \end{pmatrix} = \begin{pmatrix} 0 & 0 \\ 2e^{-j\omega\frac{T_s}{4}} & 0 \end{pmatrix}, \tag{5}$$

Equation (5) can be normalized as:

$$S' = \begin{pmatrix} 0 & 0 \\ e^{-j\omega\frac{T_s}{4}} & 0 \end{pmatrix}, \tag{6}$$

From Equation (6), the DTI can be constructed by combining the reciprocal transmission line segments with the DBG in some way. For this structure, the isolation is frequency-independent. Here, four Wilkinson power splitters (combiners) were used to connect two more reciprocal transmission lines with the DBG to construct the DTI, which is shown in Figure 3. The Wilkinson power splitters (combiners) can divide a signal into two co-phase signals with equal amplitude. The reason why they take this form as power splitters (combiners) will be explained later.

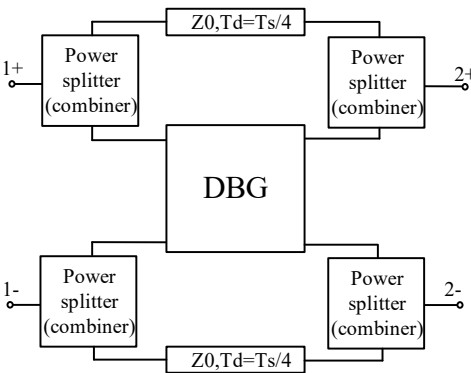

**Figure 3.** Schematic of the differential transmission isolator (DTI).

If the working bandwidth of the power divider is not considered, when a pair of differential signals input from ports 1+ and 1−, the power splitters divide the two sets of signals into four sets of signals with equivalent power. The signals which are divided from the power splitter at port 1+(1−) will pass through the upper(lower) reciprocal transmission line segment and the upper(lower) transmission line segment in the DBG, respectively. The signals will be combined at the power combiner at port 2+(2−), and no current passes through the resistors in the Wilkinson power combiners. Consequentially, the signals are not attenuated and will be output from port 2+(2−) with a delay time of Ts/4.

When the signals input from port 2+ and 2−, the signals which pass through the DBG will undergo a sign conversion. If the four signals are first combined at the power combiners at ports 1+ and 1−, the signals will not be output from ports 1+ and 1−. A current will also pass through the resistors in the Wilkinson power combiners, and therefore, the signals will get attenuated. Assuming that the signals are not absorbed by the resistors completely, the unabsorbed signals will get a virtual short and reflect back to the reciprocal transmission line segments and the DBG. Those unabsorbed signals will be combined at the power combiners at ports 2+ and 2− and be further attenuated. Moreover, the unabsorbed signals will be reflected back to ports 1+ and port 1−. Then, the signals will be combined at the power combiners in phase, and be output from ports 1+ and port 1−. However, the amplitude will have been attenuated sharply. There is no trade-off between insertion loss of front transmission

and attenuation of reverse transmission. Figure 4 illustrates the transmission process of the signals in the DTI. Similarly, the DTI is reconfigurable.

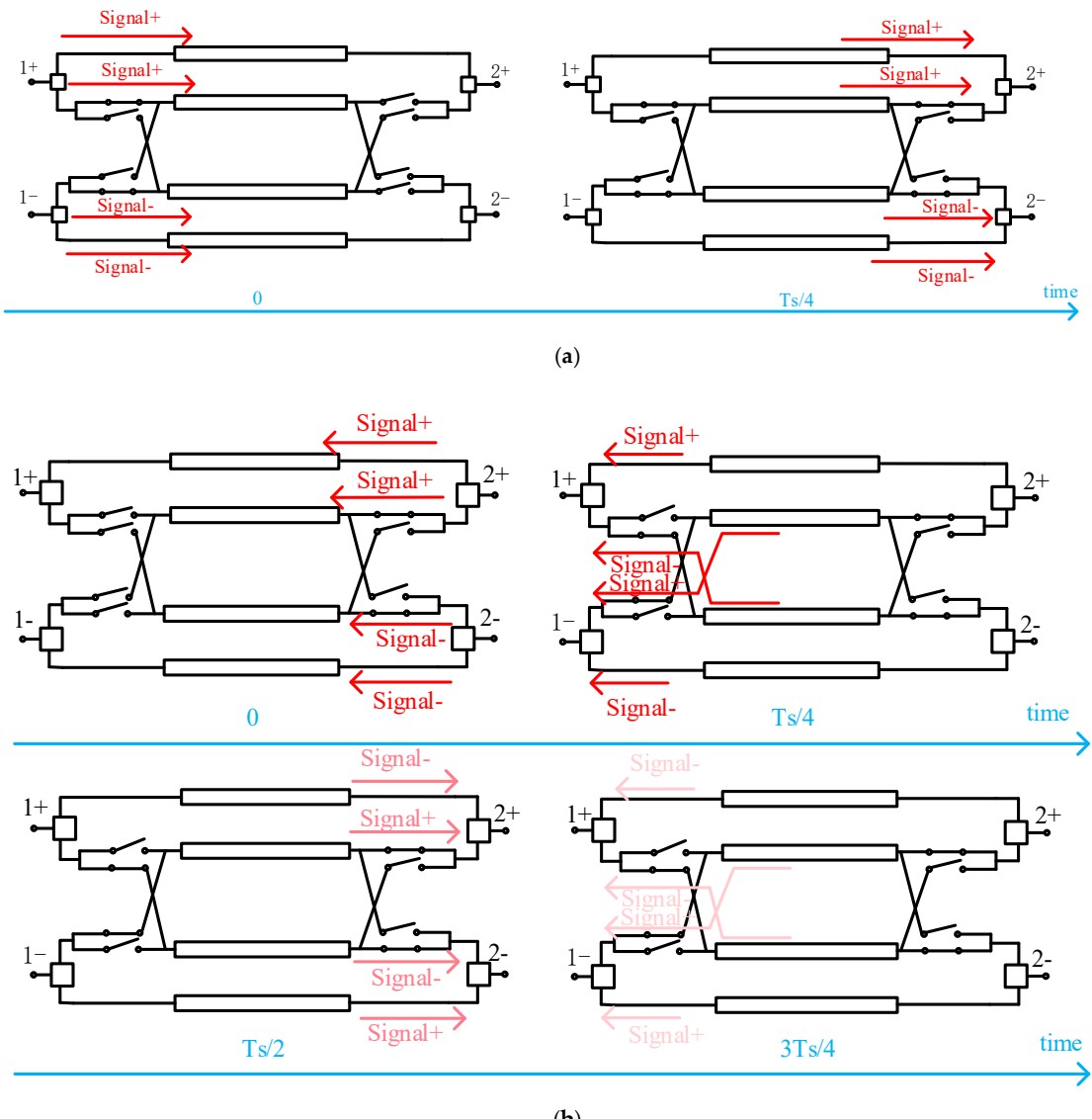

**Figure 4.** The transmission process in the DTI. (**a**) Input from port 1+(1−); and (**b**) input from port 2+(2−).

Here, the same simulation sets are used in Section 3 to verify the theory. Frequency-independent Wilkinson power splitters (combiners) were used in the simulation. The phase relationship of the control signals is the same as shown in Figure 1a. The spectrum analyzing results are shown in Figure 5. When the signals are input from ports 1+ and 1−, the signals can pass the isolator almost without loss. These simulation results are shown in Figure 5a. When the signals are input from port 2+ and 2−, there is a large loss of the signals. The isolation of the signals is about 25 dB throughout the simulation band, which can satisfy some engineering applications. Compared to the STI, the isolation is decreased by about 35 dB throughout the band.

We consider the non-ideal power splitters (combiners) in the DTI, which use the simplest structure, as shown in Figure 6, to find out the minimum operating bandwidth. For convenience, the center frequency of the power splitters (combiners) is set as 500 MHz. $Z_0$ and $\lambda_g$ are the characteristic impedance and the wavelength in the transmission lines, respectively. The simulation results from replacing the ideal power splitters (combiners) with the non-ideal power splitters (combiners) are

shown in Figure 7. Throughout the band of 100 MHz to 1 GHz, when the signals are input from ports 1+ and 1−, the insertion losses of the signals are small. They still have broadband transmission property. When the signals are input from ports 2+ and 2−, the isolation of the signals increases with the frequency. The increase ratio of the isolation is slow when the transmission frequency is close to the center frequency of the power splitters (combiners), and fast when the transmission frequency is away from the center frequency. The isolation is greater than 20 dB in the band of 400 MHz to 1 GHz. So, the non-ideal DTI also has broadband property. The operation bandwidth of the DTI can be expanded by designing broadband Wilkinson power splitters (combiners).

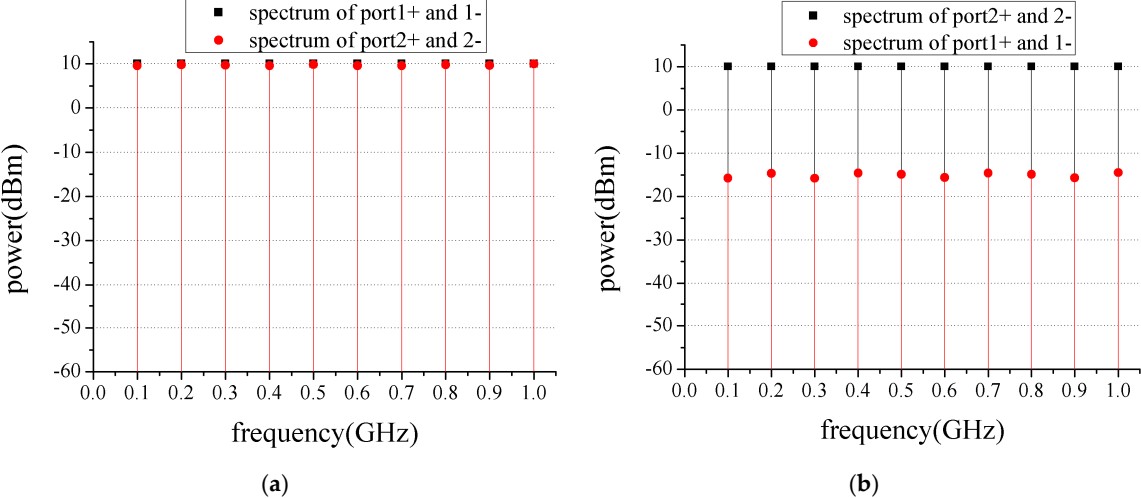

**Figure 5.** Spectrum analyzing results of the DTI. (**a**) Inputs from port 1+(1−); and (**b**) inputs from port 2+(2−).

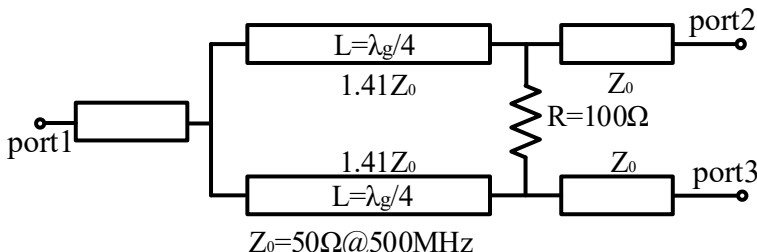

**Figure 6.** Schematic of the Wilkinson power splitters (combiners).

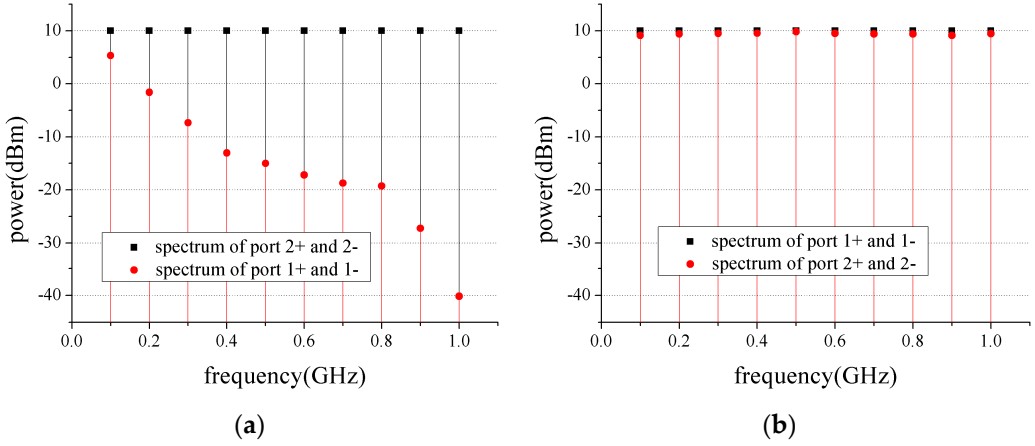

**Figure 7.** Spectrum analyzing results of the non-ideal DTI. (**a**) Inputs from port 1+(1−); and (**b**) input from port 2+(2−).

In order to verify the advantage of the DTI in anti-interference performance, a noise module was constructed, as shown in Figure 8. A noise source was added to port 1 of the STI. Moreover, a common-mode noise source was added to ports 1+ and 1− of the DTI, because the noise could be considered to be generated from a single noise source. Two frequency-independent baluns were used to connect port 1+ and port 1−, and port 2+ and port 2−, respectively. A spectrum analyzer was connected with port 2 in the STI and the balun, which connected ports 2+ and 2− in the DTI. For convenience, we assumed that the noise was a single-tone interference with a frequency of 900 MHz, and the power of the noise was set as 10 dBm. The other simulation settings were kept the same as the settings in Section 3. The simulation results of the STI are shown in Figure 8a. The results show that the noise can be output from port 2, which cannot be eliminated. However, in Figure 8b, the results show that the noise was suppressed by about 75 dB in the DTI. Similar results can be obtained for noises at other frequencies.

The main purpose of the research on magnetic-free non-reciprocal devices is to facilitate integration. One of the problems of integrated devices is that they cannot tolerate large working power due to semiconductor devices' limited power capacity ability. By using power splitters (combiners), the DTI can theoretically raise the power capacity by 6 dB as compared to the STI. Thus, the circuit can be better protected.

**The STI:**

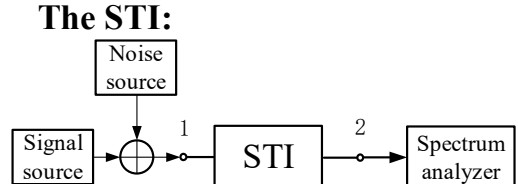

**The DTI:**

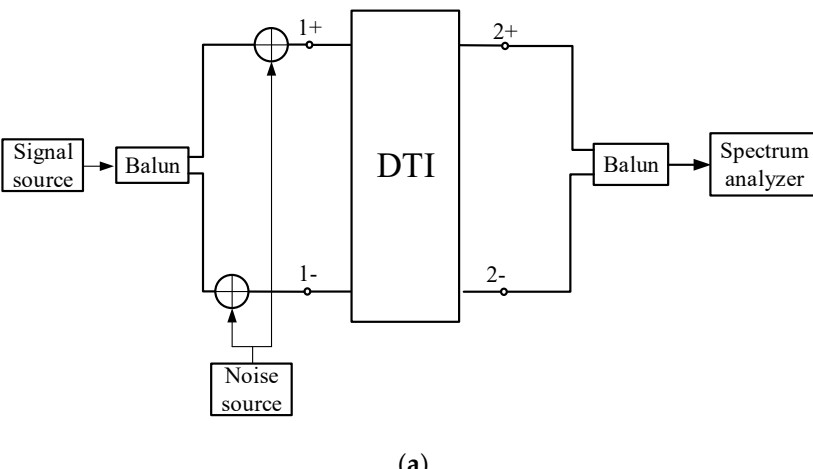

(**a**)

**Figure 8.** *Cont.*

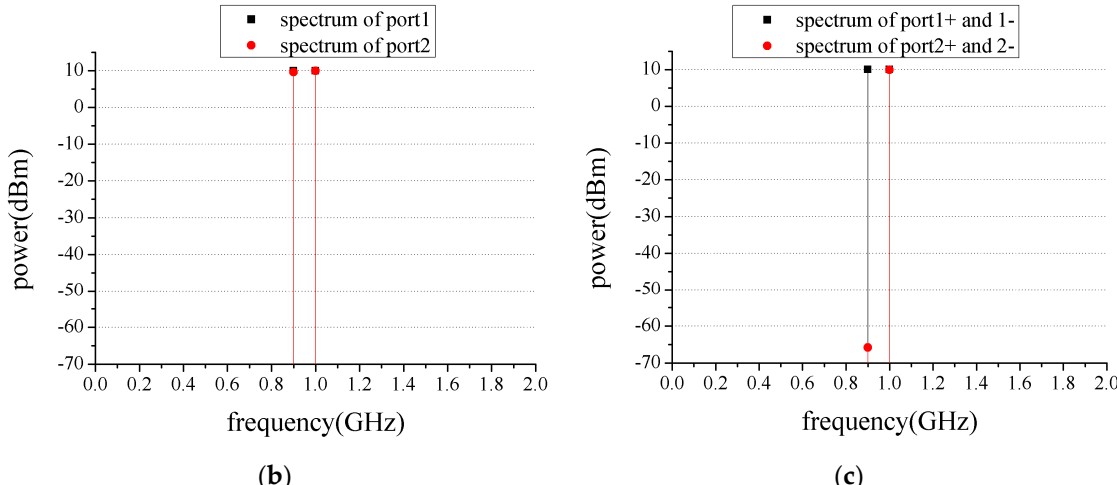

(**b**)                                                    (**c**)

**Figure 8.** (**a**) The noise module of the STI and the DTI. (**b**) Spectrum of the STI; and (**c**) spectrum of the DTI.

## 4. The Experiment of the STI

A prototype was made to verify the broadband isolation reconfigurable properties of the STI, as shown in Figure 9, which is also inspired by the previous work [14]. In this design, the GQSS in the DBG were made of a 1.6-mm FR-4 printed circuit broad (PCB) and four Minicircuits' single-pole double-throw (SPDT) switches. The two transmission line segments were realized by two 7-meter coaxial cables. The matching loads were realized by 50-Ω end caps. The four clock signals were generated by two synchronized Tektronix AFG31023C signal generators, and the frequency of the clock signals in this design was chosen as 6. 9MHz. The N5244A network analyzer was used to measure out the S parameters.

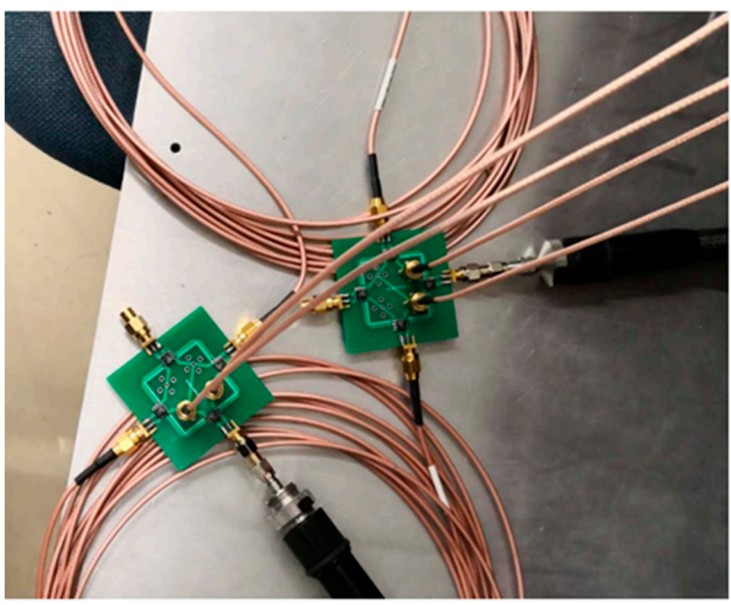

**Figure 9.** The prototype of the STI.

The measurement results are shown in Figure 10a. Throughout the measurement band of 10 MHz to 1 GHz, the return loss of the isolator is lower than −14 dB. The isolation is greater than 30 dB. The broadband property is proved. Due to the attenuation property of the coaxial cables, the insertion loss increases with the frequency, and the minimum insertion loss is 6 dB. When the S1(t) and S3(t), and S2(t) and S4(t) are exchanged, respectively, the isolation direction is changed. The measurement

results are shown in Figure 10b, and they are similar to the results of Figure 10a. The reconfigurable property is proved.

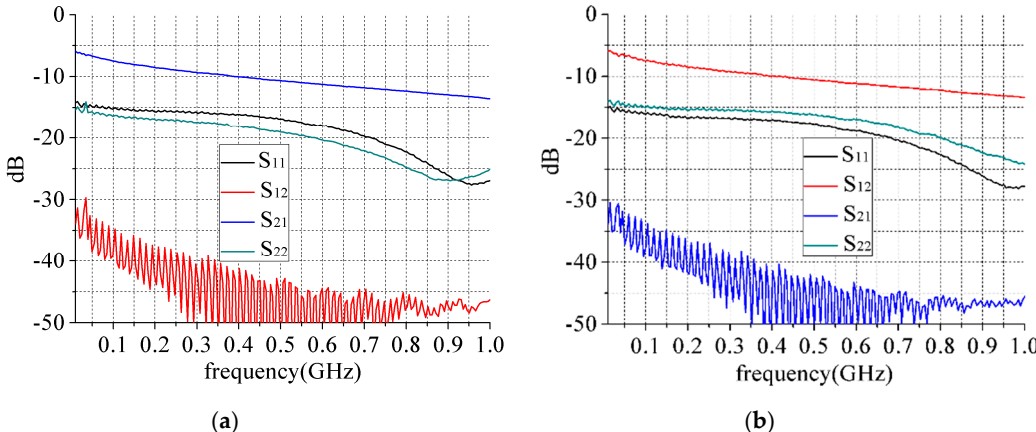

**Figure 10.** The measurement results with (**a**) clock signals unchanged and with (**b**) changed clock signals.

For comparison, the simulated and measured results of the STI are shown in Figure 11. In the observation frequency range from 100 MHz to 1 GHz, the simulated $S_{21}$ is higher than the measured $S_{21}$, and the average deviation is 12 dB. The simulated $S_{12}$ is lower than the measured $S_{12}$, and the average deviation is 21 dB. The delay error of the coaxial cables and the duty cycle error of the clock signals have significant effects on the insertion loss and return loss. The insertion loss can be reduced by using sub-nanosecond switches and high-quality clock signals.

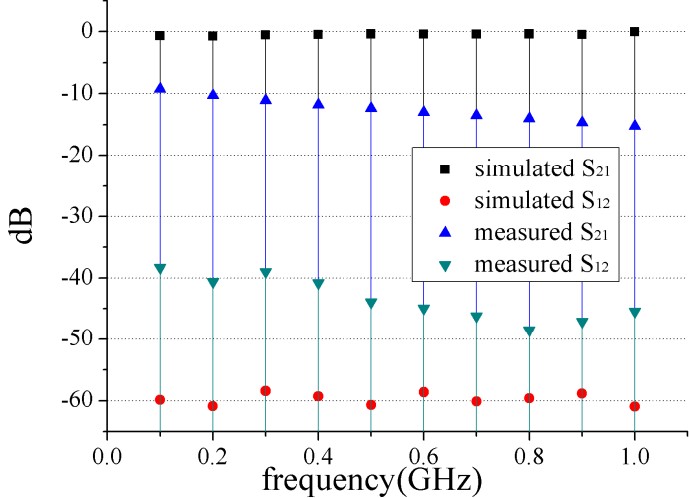

**Figure 11.** Simulation results and measurement results of the STI.

With the limitations of the experimental conditions, the delay error of the transmission line segments and duty cycle error of the clock signals could not be avoided. Thus, the signals that pass through the DBG and reciprocal transmission line segments respectively were hard to be combined with equal amplitude at the power combiners. As a result, it was difficult to verify them by using PCB circuits. The experiment of the DTI is expected to be performed in the future by using precise clock signals and advanced processing technology in the integration circuit.

## 5. Conclusions

Table 1 illustrates the properties of the four isolators. The STI has ultra-broadband and reconfigurable properties, which have not been realized at the same time by the FCI and the

UDI. Simulation and an experiment have been done to verify the structure. However, for the STI, the anti-interference ability and the power capacity ability are limited. So, the DTI has been proposed, which uses two pairs of differential signals to prevent reverse signals. The anti-interference performance can be improved by using differential transmission lines. Among the four isolators listed in Table 1, the DTI has the largest power capacity. Using the simplest Wilkinson power splitters (combiners), the DTI can also obtain a broadband property. The isolation bandwidth can be expanded by using wideband power splitters (combiners), which are easily obtained. Simulation works have been done to verify the DTI. The two kinds of magnetic-free isolators can be applied in different scenarios according to their advantages and disadvantages. The experiment of the DTI is expected to be performed in the future by using precise clock signals and advanced processing technology in the integration circuit.

**Table 1.** The properties of the isolators.

|  | STI | DTI | FCI in [13] | UDI in [13] |
| --- | --- | --- | --- | --- |
| Isolation bandwidth | Ultra-Broadband | Broadband | Limited | Ultra-Broadband |
| Reconfigurable | Yes | Yes | Yes | No |
| Anti-interference performance | Limited | Well | Well | Limited |
| Power capacity | Smallest | Largest | Medium | Smallest |

**Author Contributions:** Methodology and original draft preparation, F.W.; writing—review and editing, Y.Z. and F.Y.; supervision, Y.F.

**Funding:** This research received no external funding.

**Conflicts of Interest:** The authors declare no conflict of interest.

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
