# Peer review of "Magnetic-Free Isolators Based on Time-Varying Transmission Lines"

_electronics, doi:10.3390/electronics8060684_

Round 1
Reviewer 1 Report
This paper has proposed a single-ended transmission isolator (STI), which has a simple structure and infinite isolation bandwidth in theory, which can be easily achieved. Simulation and experiment have been done to verify the structure. But for the STI, the anti-interference ability and the power capacity ability are limited. So, a differential transmission isolator (DTI)I has been proposed to solve these problems. The results indicate the isolation of the DTI is not higher than the STI’s isolation, and the isolation bandwidth of the DTI is limited by the working bandwidth of Wilkinson power splitters (combiners). The isolation bandwidth can be expanded by using wideband power splitter (combiner), which is easy to be obtained.
The paper is very well written and results are discussed satisfactorily. However, there are few things the authors need to revise in the manuscript:
Do remove "we" from the language of the manuscript.
There are few typographical errors (e.g. in abstract showing "?"). Kindly proofread the full manuscript to remove such typos.
Show the compared result of ADS simulation against measurement results and provide the average deviation between the two.
Author Response
Thank you very much for your comment.
We have modified the manuscript and provided a response letter.
The response letter is sent as attachment.

Reviewer 2 Report
This manuscript presents magnetic-free isolations based on TVTL. This reviewer has some concerns about this work and recomends the following improvements to be made:
English language needs an in-depth review. It is hard to understand the importance of the work and to follow the explanations. Please consider to ask a native speaker to write a new version.
Since the main idea and structures of magnetic-free non-reciprocal circuits based on TVTL are already known and properly referenced, the novelty or contribution of the proposed work is not cleary stated. In the introduction it should be clearly stated the main differences of this work with respect to previously published works. Also, a comparative table before the Conclusions section would help to appreciate these main differences if any.
Theory and simulations of the presented circuits are based on frequency independent elements so the bandwidth limitations are not observed until they are evident in the Results sections. This results section is very limited and only points out to some problems that will be solved in a future work. In this reviewer's opinion, non-idealities should be considered in the simulations and a discussion should be included where the bandwidth limited results are explained, following what other authors have tried to do in referenced works such as [11].
Author Response
Thank you very much for your comment.
We have modified the manuscript and provided a response letter.
The response letter is sent as the attachment.

Round 2
Reviewer 2 Report
All my concerns about the original work have been conveniently addressed.